

# Intercomparison of total column ozone data from the Pandora spectrophotometer with Dobson, Brewer, and OMI measurements over Seoul, Korea

Jiyoung Kim[1], Jhoon Kim[1], Hi-Ku Cho[1], Jay Herman[2], Sang Seo Park[1,3], HyunKwang Lim[1], Jae-hwan Kim[4] and Koji.Miyagawa[5]

[1] Department of Atmospheric Sciences, Yonsei University, Seoul South Korea

[2] Joint Center for Earth Systems and Technology, University of Maryland, Baltimore County, UMBC-JCET and NASA Goddard Space Flight Center, Greenbelt, MD 20771, USA

[3] Research Institute for Applied Mechanics, Kyushu University, Fukuoka, Japan

[4] Department of Atmospheric Science, Pusan National University, Busan, Korea

[5] NOAA ESRL Global Monitoring Division 325 Broadway R/GMD, Boulder, CO, 80325-3328 Earth System Research Laboratory, Boulder, Colorado, USA

*Correspondence to*: Jhoon Kim (jkim2@yonsei.ac.kr)



**Abstract**
Daily total column ozone (TCO) measured using the Pandora spectrophotometer (#19) was
intercompared with data from the Dobson (#124) and Brewer (#148) spectrophotometers, as
well as from the Ozone Monitoring Instrument (OMI), over the 2-year period between March
2012 and March 2014 at Yonsei University, Seoul, Korea. The Pandora TCO measurements
are closely correlated with those from the Dobson, Brewer, and OMI instruments with
regression coefficients (slopes) of 0.95, 1.00, 0.98 (OMI-TOMS), and 0.97 (OMI-DOAS),
respectively, and determination coefficients ($R^2$) of 0.95, 0.97, 0.96 (OMI-TOMS), and 0.95
(OMI-DOAS), respectively. In particular, they show a close agreement with the Brewer TCO
measurements, with slope and $R^2$ values of 1.00 and 0.97, respectively. The difference
between the Pandora and Dobson data can be explained by smaller amount of Dobson data
available to calculate the daily averages, observation times, solar zenith angles, $SO_2$ effect,
temperature, and humidity between the two datasets. The difference in the results obtained
from the Pandora instrument and Ozone Monitoring Instrument-Differential Optical
Absorption Spectroscopy (OMI-DOAS algorithm) can be explained by the dependence on
seasonal variations of about ±2% and solar zenith angle leading to overestimation by 5% of
OMI-DOAS measurements. For the Dobson measurements in particular, the difference
caused by the inconsistency in observation times when compared with the Pandora
measurements was up to 12.5% on 22 June 2013 because of diurnal variations in the TCO
values. However, despite these various differences and discrepancies, the daily TCO values
measured by the four instruments during the 2-year study period are accurate and closely
correlated.





## 1. Introduction

Approximately 90% of total column ozone (TCO) is found in the stratosphere, and its density peak occurs at altitudes between 20 and 30 km. This layer is essential to the biosphere as it absorbs harmful solar ultraviolet (UV) radiation. In addition, UV absorption by ozone influences global radiative forcing and climate change over long timescales (e.g., Cho et al., 2003; Martens, 1998; WMO, 2014). After significant depletion of the ozone layer was detected in the 1980s (Farman et al., 1985; Chubachi, 1985), many studies have reported the recovery of the ozone hole (e.g., Harris et al., 1997; Solomon, 1999; Newchurch et al., 2003; Weatherhead et al., 2000). These studies found that the concentration of anthropogenic ozone-depleting substances (ODSs) had decreased and that, consequently, global ozone amounts would return to 1980 levels during the 21[st] century (WMO, 2014).

Over recent decades, ground-based instruments such as Dobson or Brewer spectrophotometers have been used (and improved) to obtain stable and highly accurate measurements of global ozone amounts. The Dobson spectrophotometer was developed in 1928 by G. M. B. Dobson to measure TCO levels under clear-sky conditions (Dobson, 1968). TCO values are retrieved by measuring direct or scattered intensity ratios at two wavelength pairs that have different absorption features in the UV band (A-pair: 305.5 and 325.4 nm; D-pair: 317.6 and 339.8 nm, recommended by WMO; Komhyr et al., 1980; LEONARD, 1989). Since the 1970s, many instruments have been installed and a global network established to monitor TCO amounts and its vertical profile using Umkehr measurements.

The Brewer spectrophotometer was developed in the early 1980s and since commencing operational use has supplemented measurements made by Dobson spectrophotometers





(Brewer, 1973; Kerr et al., 1985; Kerr, 2010). The measurement principle is similar to that of
the Dobson instrument; however, the Brewer spectrophotometer retrieves data on total UV
(TUV), erythemal UV (EUV), TCO, and aerosols, as well as trace gases such as $NO_2$ and $SO_2$,
by measuring solar irradiance and zenith sky radiances with an accuracy of ±1% for direct-
sun measurements. Nearly 200 Brewer instruments are now operating in about 40 countries
(Kerr, 2010), and the MK-IV version has been operating at Yonsei University, Seoul, Korea
since 1997 (Kim et al., 2014). This instrument enables measurement of global UV spectral
irradiance, and this is used for retrieval of TUV and EUV from 290 to 363 nm with a spectral
resolution of 0.5 nm on a horizontal surface (cf. Sabburg et al., 2002). It also measures
normal direct UV radiation, which can be used to retrieve gas concentrations at five
wavelengths in the UV region (306.3, 310.1, 313.5, 316.7, and 320.0 nm; e.g., Kerr et al.,
1985; Kerr, 2002; Kim et al., 2011). In addition, satellite-based observations, such as Total
Ozone Mapping Spectrometer (TOMS) and OMI, have also been conducted since 1979
(Bhartia and Wellemeyer, 2002; Levelt et al., 2006) and have generated an extensive and
highly accurate global dataset. These data have been validated globally and over long periods,
and have been widely used in numerous studies (Balis et al., 2007; McPeters et al., 2008;
WMO, 2014).
The Pandora spectrophotometer was developed at NASA's Goddard Space Flight Center in
2006 to measure the concentrations of trace gases including ozone (Herman et al., 2009; Cede,
2011). It consists of a head sensor with fore-optics, mounted on a high-precision sun-tracker
and sky-scanner (ca. 1.6° field of view and ca. 0.01° pointing precision), and it measures the
direct solar beam in the spectral range between 280 and 500 nm using the Sun-only CMOS
detector, and 280–525 nm using the Sun-and-Sky CCD detector with absolute $O_3$ retrieval





errors of about 1% (±3 DU) and a high precision of ±0.1 DU (Herman et al., 2015; Reed et al.,
2015; Tzortziou et al., 2012). Absolute $NO_2$ retrieval errors are about ±0.1 DU (Herman et al.,
2009). From the measured radiance, TCO levels, together with the total column of trace gases
(including $NO_2$, $SO_2$, BrO, water vapor, and formaldehyde), are retrieved using the
differential optical absorption spectroscopy (DOAS) technique (Wang et al., 2010; Yun et al.,

74    2013).

In this study, we intercompare the Pandora measurements from Seoul with two ground-based
and two satellite datasets over a 2-year period. Furthermore, the difference between Pandora
and the other measurements, and the causes of these differences, are discussed. The
remainder of this paper is organized as follows. Section 2 describes the ground-based and
satellite datasets used in this study. Section 3 describes the methodology and results of the
intercomparison together with our analysis and discussion. In addition, high-resolution
diurnal variations in the Pandora TCO data are compared with Dobson measurements. Finally,
our conclusions are summarized in Sect. 4.











## 2. Data and Analysis

In this study, the TCO data used for intercomparisons were measured using Pandora, Dobson, and Brewer spectrophotometers from March 2012 to March 2014 at Yonsei University (37.57°N, 126.95°E; 84 m above sea level) in Seoul, Korea. The university is one of the WMO Global Ozone Observing System (GO3OS) stations (Station No. 252). An OMI has also recorded TCO data over this site since 2004. As part of the ongoing national monitoring program of the Korea Meteorological Administration (KMA), TCO measurements have been made at this station since 1984. The calibration history and characteristics of Dobson (Beck #124), Brewer (SCI-TEC #148), and OMI instruments are described in Sect. 2.1 to 2.4.

## 2.1. Dobson Spectrophotometer (Beck #124)

The Dobson spectrophotometer (Beck #124) is located on the rooftop of the Science Hall of Yonsei University and has been in operation since 1984, with regular calibration as a standard for total ozone measurements (Cho, 1989, 1996; Cho et al., 2003; Kim et al., 2005). The instrument retrieves TCO from the observed UV radiance in direct-sun and zenith-sky modes three times a day. A direct-sun TCO value measured at noon under clear skies is generally selected as a representative value; however, a value close to noon or the zenith-sky measurement can be used instead if data from noon are unavailable. After the automation of the Dobson instrument (in particular, Q-levers, Attenuator, R-dial, observation, and data processing with test) in 2006, accuracy was improved such that the proportion of data points within a ±3% error range increased from 92% to 98% (Kim et al., 2007; Miyagawa et al., 2005). The calibration history of this instrument has been summarized by Kim et al. (2007)





and Hong et al. (2014). The Dobson instrument has provided a high-quality, objective, and
reliable dataset that can be used to monitor the variations and trends in ozone levels over the
Korean Peninsula. According to previous studies that have used this dataset, the annual mean
ozone level increased by 7.2% decade$^{-1}$ from 2004 to 2010, whereas from 1979 to 2004 it
decreased slightly by 0.41% decade$^{-1}$ (Kim et al., 2005, 2014).

**2.2. Brewer Spectrophotometer (SCI-TEC #148)**
The Brewer MK-IV spectrophotometer (SCI-TEC #148) at the Dobson measurement site,
which has been in operation since 1997 (Kim et al., 2011), automatically measures TCO,
trace gases, and UV irradiance, and is regularly calibrated (Kim et al., 2014). Previous studies
based on Brewer spectrophotometer data have shown that annual EUV and TUV from 2004
to 2010 decreased by 0.83% decade$^{-1}$ and 0.90% decade$^{-1}$, respectively (Kim et al., 2011),
whereas the Aerosol Optical Depth (AOD) at 320 nm increased by 22.4% decade$^{-1}$ (Kim et
al., 2014). And Hong and Cho (2007) showed the annual mean variation of the daily total
ozone amount showing a clear seasonal variation in Pohang from 1994 to 2005 using Brewer
spectrophotometer.

**2.3. Ozone Monitoring Instrument (OMI)**
The OMI onboard the Aura satellite has been dedicated to monitoring ozone and trace gases
for air quality and climate studies since 2004. This instrument detects the molecular
absorption of backscattered solar light in the visible and UV wavelengths (270–500 nm) with



a spatial resolution of 13 × 24 km at nadir (Buchard et al., 2008; Levelt et al., 2006). The total
ozone product from the OMI is calculated using two different algorithms: the TOMS
algorithm and the DOAS algorithm, which show fairly good agreement, with correlation
coefficients ranging from 0.89 to 0.99 (Antón et al., 2009; Kroon et al., 2008; McPeters et al.,
2008). The TOMS algorithm uses two wavelengths: a weak absorption wavelength (331.2 nm)
and a strong absorption wavelength (317.5 nm) to derive TCO. The derivation applies tables
calculated by the TOMS forward model (TOMRAD), which is based on successive iterations
of the characteristic equation in the theory of radiative transfer developed by Dave (1964)
(Bhartia and Wellemeyer, 2002; McPeters et al., 2008). The DOAS algorithm derives TCO
using the DOAS method, and the derivation follows three steps. First, spectral fitting is
performed (Platt and Stutz et al., 2008; Veefkind et al., 2006) to obtain the slant column
density (SCD); second, the SCD is converted to the vertical column density (VCD) using the
air mass factor (AMF). The final step in the derivation procedure is a correction for the cloud
effect (e.g., Hong et al., 2014; McPeters et al., 2008). For the level-3 daily product used in
this study, this step consists of gridding and averaging the level-2 TCO orbital swath data
onto the 0.25° × 0.25° global grids (after a quality check).

**2.4. Pandora Spectrophotometer (#19)**
The Pandora spectrophotometer (#19) was installed at Yonsei University as part of the
Distributed Regional Aerosol Gridded Observation Networks (DRAGON)-NE Asia
Campaign                                        (http://aeronet.gsfc.nasa.gov/new_web/DRAGON-
Asia_2012_Japan_South_Korea.html) and has been used operationally since March 2012 to



monitor $NO_2$ and $O_3$ concentrations. Pandora is a passive system that measures direct sunlight
from 280 to 525 nm with a spectral resolution of 0.6 nm, and uses a UV sensitive CCD
detector of $2048 \times 16$ pixels. Two UV band-pass filters, BP300 (280–320 nm) and U340
(280–380 nm), are used to correct for the stray light effect. The temporal resolution of the
Pandora measurement is about 2 minutes (Tzortziou et al., 2012; Yun et al., 2013), and it was
installed at the same geolocation as the Dobson and Brewer instruments.















**3. Results**
The daily TCO datasets were calculated using the following methods. Real-time Pandora and
Brewer data obtained from the direct-sun measurements were averaged to obtain a single
representative daily value. For the Pandora data in particular, to avoid errors associated with
cloud contamination and stray light effects, the data were selected using the following criteria:
root mean square (RMS) < 0.05, solar zenith angle < 75°, and uncertainty of ozone amount <
2 DU, as suggested in previous studies (Herman et al., 2015; Reed et al., 2015; Tzortziou et
al., 2012). For the Dobson spectrophotometer, daily values measured in direct-sun mode
under a clear sky were averaged for the comparison. Finally, for the OMI instruments, the
daily level-3 gridded data, the OMTO3e from the TOMS-like algorithm, and the
OMDOAO3e from the DOAS technique, together with site information from Yonsei
University, were spatially interpolated.

**3.1. TCO measured by the Pandora, Dobson, Brewer, and OMI instruments**
The time series of measurements from the four instruments are shown in Fig. 1 for
comparison, and all fall within the range of 230–500 DU and show obvious and similar
seasonal variations. These seasonal variations are caused by changes to the Brewer–Dobson
circulation in the Northern Hemisphere (Brewer, 1949; Wang et al., 2015; Weber et al., 2003).
Additionally, Fig. 2 shows similar annual cycles with an amplitude of about 0.15% of the
average values for the four different instruments. Maximum and minimum values of 2-year
averaged monthly TCO and annual ranges are also recorded in this figure. All graphs and
statistics were derived from Fig. 1 under the condition that the valid number of daily





observations was greater than 10 days per month. In this figure, the largest maximum
monthly mean TCO values are from the Dobson measurements (i.e., 371.5 DU in April) and
the smallest minimum monthly mean TCO values were from the Pandora measurements (i.e.,
268.9 DU in October). In addition, the largest annual range is seen in the Dobson
measurements (101.7 DU), whereas the smallest range is seen in the Brewer measurements
(81.3 DU). In addition, the maximum monthly mean TCO value of the Pandora
measurements shows the smallest relative difference with that of the Brewer measurements of
1.54% and the minimum monthly mean TCO value of the Pandora measurements shows the
smallest relative difference with that of the Dobson measurements of 0.37%. The OMI-
DOAS measurements also showed the smallest difference in minimum value from that of the
Pandora of 0.37%. Next, we consider the daily data (Table 1).
Table 1 shows the average, standard deviation, and maximum and minimum values of the
daily TCO data measured by the four instruments, together with the relative differences
between the Pandora data and the Dobson, Brewer, and OMI data. The largest maximum and
smallest minimum TCO values were 467.1 DU on 10 April 2013 and 238.3 DU on 8 October
2013, respectively, measured by OMI-TOMS. For the 2-year average TCO value, the Dobson
value was the largest at 331.9 DU and with a standard deviation of 38.6. In contrast, the
Pandora measurement showed the smallest two-year average value of 317.2 DU, with a
standard deviation of 36.8 and a maximum of 436.7 on 6 April 2012, and the smallest
minimum value of 249.2 DU on 7 October 2013. The histograms of all daily TCO data (Fig.
3) show a generally Gaussian distribution and suggest that the 2-year average values of the
Pandora, Dobson, Brewer, and OMI instruments in Table 1 are a reliable representation of the
annual mean TCO value over the 2-year period for each instrument. The annual mean TCO





values from 2012 to 2014 are largest for the Dobson instrument (331.9 ±38.6 DU) and
smallest for the Pandora instrument (317.2 ±36.8 DU). The annual and monthly mean TCO
values over this period are similar to those of the Dobson unit from 1985 to 2000 (within
~2%), as presented by Cho (2003). However, slight decreases in the annual and monthly
means of the TCO values are seen over our study period (2012–2014) when compared with
the earlier period (1985–2000) for Pandora (decreases of –1.49% and –0.54%, respectively);
the Dobson, Brewer, and OMI show slight increases of ~1%. Statistical details are presented
below.

**3.2. Intercomparisons of Pandora TCO measurements**
In this study, the linear-least-squares regression method was used for all intercomparisons. To
ensure high reliability of intercomparison results, only datasets valid for all instruments were
selected. To this end, prior to making the comparisons it was necessary to verify the accuracy
of the datasets. Thus, intercomparisons of all available TCO values obtained from each
instrument (except for Pandora) were performed for the study period. As illustrated in Fig. 4,
all of the regressions show excellent agreement, with slopes close to 1:1 and coefficient of
determination ($R^2$) greater than 0.90. In particular, the Brewer data show results close to a
perfect correlation with those of the Dobson and OMI instruments, with slopes of 1.01 and
0.95, and $R^2$ values of 1.00 and 0.97 (OMI-TOMS), respectively. These strong correlations
among the datasets indicate reliable measurements with high accuracies, thereby enabling a
thorough regression analysis. Having established the reliability of the datasets, we next
carried out the intercomparisons of the Pandora TCOs.





Figure 5 shows scatter plots of the daily Pandora TCOs and daily Dobson, Brewer and OMI
TCOs, respectively, together with regression lines within an error range of ±3% (Basher,
1985; Tzortziou et al., 2012) calculated by linear least-squares regression. The slope of the
regression line and the coefficient of determination ($R^2$) from the intercomparison of the
Pandora data with the other instruments are 0.95 and 0.95 for the Dobson data, 1.00 and 0.97
for the Brewer data, 0.98 and 0.96 for the OMI-TOMS data, and 0.97 and 0.95 for the OMI-
DOAS data, respectively. That is, all linear regression lines between Pandora and the others
show best fit. Furthermore, the Pandora data show the highest mean ratio value of 0.98±0.001
(±1σ) with the Brewer data, which is slightly higher than the others. According to Park et al.
(2012), the mean ratio value shows intercomparison accuracy. These high correlation results
are comparable with previous validation studies undertaken in Boulder, Colorado (Herman et
al., 2015) and in Greenbelt, Maryland (Tzortziou et al., 2012). Table 1 lists the mean relative
differences, which are defined as the percentage differences between the observation data. All
of these values show that the measured TCO values from the Dobson, Brewer, and OMI
instruments are generally higher than those from Pandora. Figure 6 shows time series of the
relative differences between the daily TCO values from Pandora and the other instruments,
which is smallest between the OMI-DOAS and Pandora data (2.01% on average), but
increases to 3.64%, 2.31%, and 2.55% for the Dobson, Brewer, and OMI-TOMS data,
respectively. Based on these results, we conclude that the Dobson, Brewer, and OMI TCO
data show good agreement with the Pandora measurements.
We used a generic analysis of variance table for simple linear regression (ANOVA) to
perform a more detailed analysis of these relationships. ANOVA tables for the Pandora
intercomparisons are presented in Table 2 and follow the procedure of Wilks (2006). Table 2





((a)-(d)) shows the mean squared error (MSE), standard error (s.e.), and F-ratio of the
Pandora intercomparison results. The MSE indicates the variability of the data, with a large
MSE indicating a greater degree of scatter around the 1:1 line, and a small MSE the opposite.
The MSE value for the comparison of Pandora with Brewer was the smallest, at 42.8, and
was the largest (73.7) with OMI-DOAS. That is, in the case of comparison between the
Pandora and the Brewer, most of data are located close to the regression line (Fig. 5). The s.e.
of the slope and the intercept represent the uncertainty of the regression line. The s.e. value of
the slope was 0.02 for all Pandora intercomparisons, and that of the intercept was the smallest
(5.42) and largest (6.95) for the comparisons with the Brewer and OMI-DOAS instruments,
respectively. Finally, the F-ratio (regression mean square (MSR)/MSE) increases with the
strength of the regression (Draper et al., 1966; Neter et al., 1996). In Table 2, the F-ratio value
calculated from the regression analysis of Pandora with the Brewer is 3477.9, which is much
greater than the others (2351.5, 2607.4, and 1974.8 for Dobson, OMI-TOMS, and OMI-
DOAS, respectively). Taking all of these results into consideration, the TCO data measured
by Pandora are in closest agreement with the Brewer data, similar to the validation results of
Tzortziou et al. (2012).
The relatively small slopes, $R^2$, and F-ratios, and large MSE show that the Pandora data have
a slightly weaker linear relationship with the Dobson and OMI-DOAS data than with the
Brewer and OMI-TOMS data (Fig. 5; Table 2). In particular, in the case of OMI-DOAS, the
regression result shows the smallest $R^2$ and F-ratio values of 0.95 and 1974.8, respectively,
and the largest MSE of 73.7, even though it has the smallest mean relative difference of
2.01%. However, the time series of relative differences between the Pandora and OMI-DOAS
TCO data in Fig. 6(d) shows more negative relative difference values than for the other



relationships, and these compensate for the positive values. That is, for OMI-DOAS, there are
more and larger underestimated TCO values when compared with the Pandora data than for
the Dobson, Brewer, and OMI-TOMS data and these underestimated TCO values lead to the
small mean relative difference. As a result, it is difficult to conclude that the Pandora and
OMI-DOAS TCO values are in good agreement only with small mean relative difference
value between two data. Moreover, the largest MSE and smallest F-ratio values, which are
used to assess the correlation between the Pandora and OMI-DOAS data, represent a poorer
agreement among all intercomparison results with an MSE of 73.7 and F-ratio of 1974.8
(Table 2(d)). Thus, in summary, the Pandora TCO data show a better correlation with the
Brewer or OMI-TOMS data than do the OMI-DOAS data. This result can be explained by the
dependence of the OMI-DOAS measurements on seasonal variations and solar zenith angle.
According to previous studies, for a comparison between ground-based and OMI instruments,
OMI-DOAS TCO data have a seasonal variation of about ±2% and can be overestimated by 5%
depending on solar zenith angle (Balis et al., 2007; Kroon et al., 2008; McPeters et al., 2008).
The Pandora TCO values show very good agreement with the Dobson values, with a slope of
0.95 and $R^2$ of 0.95. This result is similar to the findings of Herman et al. (2015), despite the
following error sources in the Dobson measurements:
• the limited amount of data used to calculate the single representative daily
average;
• the dependence on solar zenith angle, meaning that measurements are
underestimated by 6% or overestimated by 20%−30% when solar zenith angles are
less than 57° and greater than 60°, respectively (Bojkov, 1969; Komhyr, 1980;
Miyagawa et al., 2005; Nichol and Valenti, 1993);



• the $SO_2$ absorption effect (De Backer and De Muer, 1991; Komhyr, 1980;
Miyagawa et al., 2005); and
• fixed temperature and high humidity lead to a bias in TCO retrievals
(Herman et al., 2015; Komhyr, 1980).
According to Herman et al. (2015), both the standard Dobson and Pandora TCO retrievals
required a correction using a monthly varying effective ozone temperature for removing
seasonal bias.

### 3.3. Diurnal variations in Pandora TCO

As mentioned above, the temporal resolution of the Pandora measurements is about 2 minutes,
and this allows us to detect diurnal variations in ozone using the Pandora data. Figure 7
shows six cases of diurnal variation for the TCO values measured by the Pandora instrument
with average, minimum, and maximum values under clear-sky condition when the cloud
amount is less than 3 tenths during the study period. In this figure, TCO data measured at
solar zenith angles greater than 75° are shaded and excluded from the statistical calculations.
According to Herman et al., (2015), the Pandora (#34) TCO data measured at Boulder,
Colorado over 13 consecutive days in December 2013 showed considerable variations.
Similarly, in Fig. 7 there are substantial daytime variations for all six cases, especially on 5
March 2013 (Fig. 7 (c)), which shows the largest standard deviation of 15.4 DU. Moreover,
the range of TCO values on a given day shows a largest value of 53.4 DU, about 15.3% of the
daily average value. Because of these variations, the inconsistency of time intervals between
measurements selected for the daily averaging in the intercomparison can result in a sampling





bias. In particular, direct-sun observations by the Dobson instrument were performed, at most,
three times a day in this study. Observation times and real-time TCO values, as well as the
daily average values of the Dobson measurements, are shown for each diurnal cycle in Fig. 7.
In the six cases, the daily TCO values from the Pandora instrument were underestimated by
about 5% compared with those of the Dobson. For the entire period, the maximum difference
between the daily TCO values of the Pandora and Dobson was ~12.5% on 22 June 2013.
Thus, Herman et al. (2015) suggested that the Pandora time interval for intercomparison with
Dobson should be kept fairly short less than 8 minutes to avoid under-sampling of the
coincident time series. More reliable characteristics of diurnal variability of TCO can be
found using the long-term Pandora measurement data in the future.














### 4. Summary and Conclusions

In this study, daily total ozone measured by the Pandora spectrophotometer were intercompared using ground-based and satellite measurements (Dobson and Brewer spectrophotometers, and OMI) over a 2-year period at Yonsei University, Seoul, Korea. A linear least-squares regression analysis revealed that the Pandora TCO data show excellent agreement with the other instruments, with slopes close to 1 and $R^2$ values greater than 0.95, which are within ±5% of perfect regression. In addition, comparison of the mean relative differences shows that the Pandora TCO data were underestimated when compared with the Dobson, Brewer, and OMI data. Through detailed comparison using the ANOVA approach, we found that the regression of the Pandora intercomparison with the Brewer data shows the smallest MSE value of 42.8 and the largest F-ratio of 3477.9, indicating a close relationship. Several internal and external factors may result in slight differences between the Pandora measurements and other data; i.e., the time interval difference for daily averaging, dependence on solar zenith angle, $SO_2$ effect, temperature, and humidity for the Dobson data, and dependence on seasonal variations and solar zenith angle for the OMI-DOAS data. In particular, the Pandora measurements were underestimated by up to about 12.5% compared with the TCO obtained from the Dobson instrument on 22 June 2013. Despite these factors, daily TCO values retrieved from Pandora showed very good agreement with the Dobson, OMI-DOAS, Brewer, and OMI-TOMS data. Consequently, daily total ozone data measured by the Pandora spectrophotometer show high reliability, and are expected to improve substantially with the regular and accurate calibration and validation associated with the operational monitoring of trace gases and pollutants.



*Acknowledgements.* This research was supported by the GEMS program of the Ministry of
Environment, Korea and the Eco Innovation Program of KEITI (2012000160002) and the
KORUS AQ program of the National Institute of Environmental Research(NIER) and Korea
Aerospace Research Institute(KARI).



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





**Table 1.** Summary of intercomparison statistics for the 2 years from March 2012 to March 2014..

|  | **Pandora** | **Dobson** | **Brewer** | **OMI-TOMS** | **OMI-DOAS** |
|---|---|---|---|---|---|
| Average [DU] | 317.2 | 331.9 | 325.1 | 324.1 | 322.0 |
| Standard deviation | 36.8 | 38.6 | 36.2 | 38.0 | 38.6 |
| Max (date) | 436.7 (12/Apr/6) | 463.0 (13/Apr/26) | 449.3 (13/Apr/26) | 467.1 (13/Apr/10) | 465.1 (13/Apr/10) |
| Min (date) | 249.2 (13/Oct/7) | 239.0 (13/Oct/7) | 246.5 (13/Oct/7) | 238.3 (13/Oct/8) | 241.8 (13/Oct/8) |
| **Mean relative difference [%]** | | | | | |
| Dobson–Pandora | Brewer–Pandora | | OMI-TOMS–Pandora | | OMI-DOAS–Pandora |
| 3.64 | 2.31 | | 2.55 | | 2.01 |





**Table 2(a).** ANOVA table for simple linear regression between the Pandora and Dobson data.[1] .

| Source | df | SS | MS | F | P |
|---|---|---|---|---|---|
| Total | 114 | 153,818 | | | |
| Regression | 1 | 146,765 | 146,765 | 2351.5 | < 0.0001 |
| Residual (error) | 113 | 7053 | 62.4 | | |
| | | | | | |
| Variable | *Coefficient* | *s.e.* | *t ratio* | | |
| Intercept | 5.21 | 6.35 | 0.82 | | |
| Slope | 0.95 | 0.02 | 48.5 | | |


**Table 2(b).** As for Table 2(a) but for comparison of the Pandora and Brewer data.

| Source | df | SS | MS | F | P |
|---|---|---|---|---|---|
| Total | 114 | 153,818 | | | |
| Regression | 1 | 148,978 | 148,978 | 3477.9 | < 0.0001 |
| Residual (error) | 113 | 4840 | 42.8 | | |
| | | | | | |
| Variable | *Coefficient* | *s.e.* | *t ratio* | | |
| Intercept | −6.15 | 5.42 | −1.14 | | |
| Slope | 1.00 | 0.02 | 59.0 | | |


---

[1]  The column headings df, SS, MS, F, and P stand for degrees of freedom, sum of squares, mean square, F-ratio, and P-value, respectively.



**Table 2(c).** As for Table 2(a) but for comparison of the Pandora and OMI-TOMS data.

| Source | df | SS | MS | F | P |
|---|---|---|---|---|---|
| Total | 114 | 153,818 | | | |
| Regression | 1 | 147,429 | 147,429 | 2607.4 | < 0.0001 |
| Residual (error) | 113 | 6389 | 56.5 | | |

| Variable | *Coefficient* | *s.e.* | *t ratio* |
|---|---|---|---|
| Intercept | −1.66 | 6.17 | −0.27 |
| Slope | 0.98 | 0.02 | 51.1 |



**Table 2(d).** As for Table 2(a) but for comparison of the Pandora and OMI-DOAS data.

| Source | df | SS | MS | F | P |
|---|---|---|---|---|---|
| Total | 114 | 153,818 | | | |
| Regression | 1 | 145,493 | 145,493 | 1974.8 | < 0.0001 |
| Residual (error) | 113 | 8325 | 73.7 | | |

| Variable | *Coefficient* | *s.e.* | *t ratio* |
|---|---|---|---|
| Intercept | 4.46 | 6.95 | 0.64 |
| Slope | 0.97 | 0.02 | 44.4 |










 

**Figure 1.** Daily TCO values from the following instruments: (a) Pandora, (b) Dobson, (c) Brewer, (d) OMI-TOMS, and (e) OMI-DOAS, for the 2 years from March 2012 to March 2014.






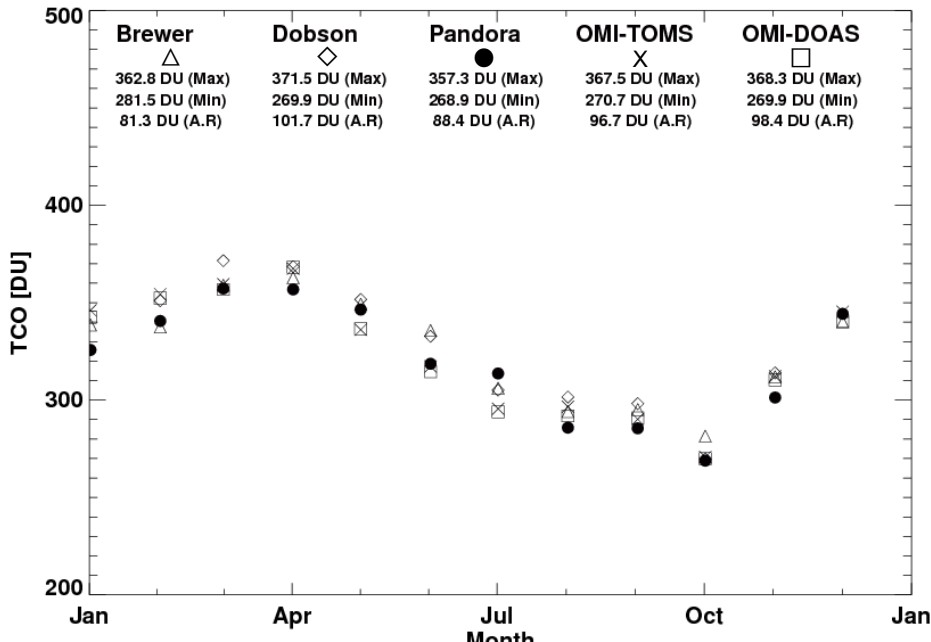


**Figure 2.** 2-year averaged monthly TCO values, together with the maximum, minimum values and annual
ranges (A.R) from the four instruments over the study period.





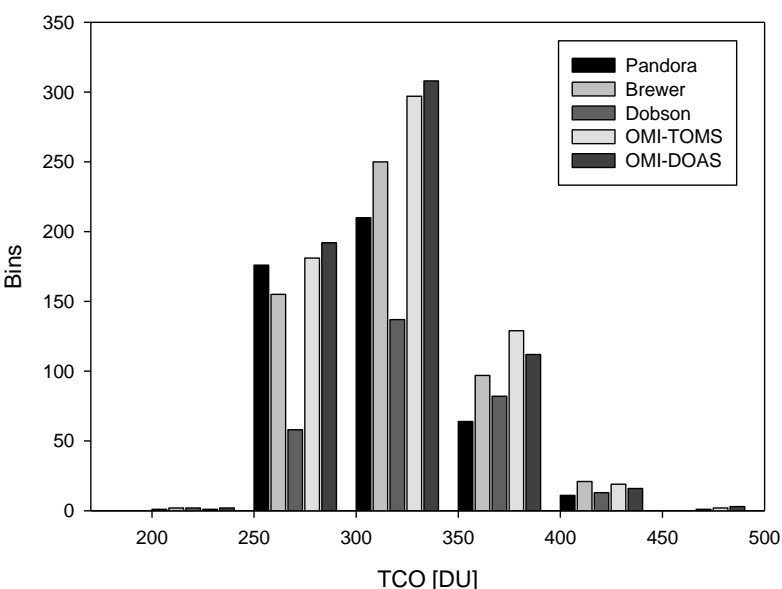


**Figure 3**. Histogram of daily TCO values from the four instruments (Pandora, Brewer, Dobson, and OMI (TOMS and DOAS).
























**Figure 4.** Intercomparison of daily TCO values between (a) Dobson and OMI-TOMS, (b) Dobson and OMI-DOAS, (c) Brewer and OMI-DOAS, (e) Brewer and Dobson, and (f) OMI-DOAS and OMI-TOMS.








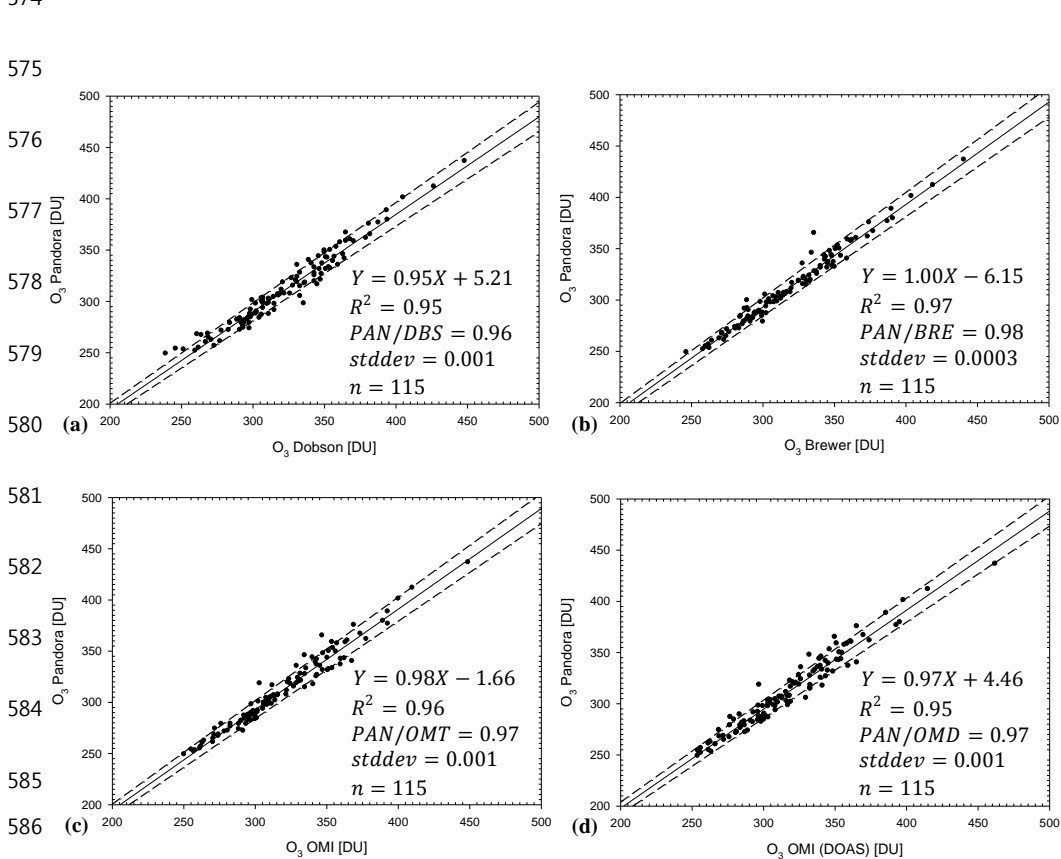

**Figure 5.** Intercomparison of daily TCO values from Pandora with (a) Dobson, (b) Brewer, (c) OMI-TOMS, and (d) OMI-DOAS. Black lines represent regression lines and blue lines indicate an error range of ±3%.





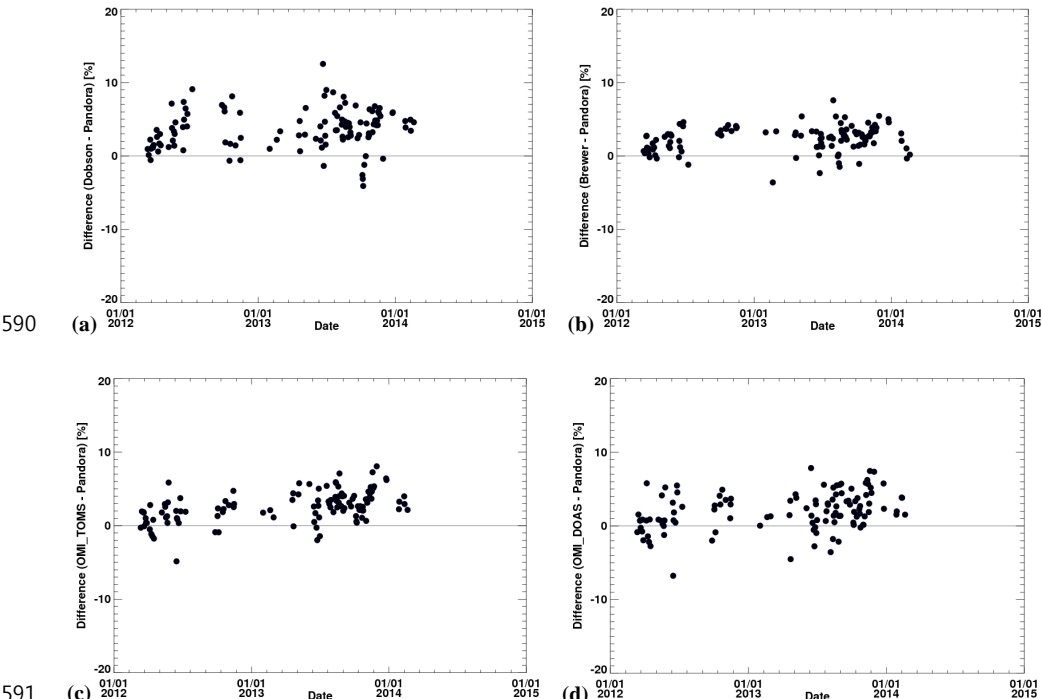

(a)
(c)
**Figure 6.** Time series of relative differences in daily TCO values from Pandora and those from (a) Dobson, (b)
Brewer, (c) OMI-TOMS, and (d) OMI-DOAS ($\frac{TCO_{inst} - TCO_{Pan}}{TCO_{Pan}}$ [%]). Gaps in the time series indicate at least one
null value in the observations from the four instruments.



(a)
(c)
(e)

**Figure 7.** Diurnal variations in TCO values retrieved from Pandora for six randomly selected clear-sky days (cloud amount < 3) during the study period, on (a) 5 April 2012, (b) 15 October 2012, (c) 5 March 2013, (d) 3 June 2013, (e) 27 January 2014, and 11 February 2014. TCO values measured at solar zenith angle > 75° are shaded and were removed from the calculations. Filled circles and dashed lines represent direct-sun TCO values measured by the Dobson instrument and observation times, respectively. All vertical axes have the same scale-range of 100 DU.