# Peer review of "Intercomparison of total column ozone data from the Pandora spectrophotometer with Dobson, Brewer, and OMI measurements over Seoul, Korea"

_Atmospheric Measurement Techniques, 2016_

## Referee Comment (RC1) · Anonymous Referee #1 · 5 Aug 2016

The paper compares the measurement of total ozone colums measured with the relatively new Pandora instrument with data from other techniques at a specific site. Although similar comparisons have already been published it is of scientific value to have this aditional comparison to evaluate the quality of the data obtained with the different techniques.

The manuscript is well structured but at some placed the explanations could be more precise (see specific comments).

In section 3.3 the diurnal variations are discussed, but only in conjunction with the

[Figure]

Dobson instrument. It would be interesting to include a comparison of the diurnal variations with the Brewer instrument which, I suppose has more observations per day than the Dobson, and which shows the best correlation with the Pandora.

Specific comments:

line 21-22: "... are accurate and closely correlated" this is vague and could be more precise from the results.

line 48: the type of observations possible with a Brewer instrument depend on the version of the instrument

line 49: please reformulate with a separate sentence to make clear that the 1% accuracy holds for direct sun observations only

line 112: Although references are given about the calibration history it would be of interest to the reader to mention the calibrations (and the conclusions) that are relevant for the period of the comparison.

line 115: a trend from 1979 to 2004 is mentioned while in line 102 it is mentioned that the instrument became operational in 1984. Please clarify

In the section starting at line 118: please mention, as for the Dobson instrument, the calibrations (and their conclusions) that are relevant for the comparison period.

line 179: please specify which RMS is meant (of the observations during the day?) and what is meant by uncertainty of ozone amout

line 201-204: reformulate (split sentence).

line 247-248: what is meant by "all () lines () show best fit"?

line 273 : is MSR abreviation for "mean square regression"?

line 303 and following: what would be the estimated effect of each of the possible error sources (eg what is the SO2 effect at Seoul?)

Technical comments:

Figure 1 : the 5 panels could be combined in one larger panel, with the different instruments represented in different colors/symbols. The same colors/symbols could then be used in Figure 2

Figure 3: if colors are used in fig s 1 and 2, the same colors can be used in fig 3. please mention what is the with of the bins of the histogram?

Caption fig 6 (line 594): replace "null" by "missing"

―――――――――――――――――

---

## Referee Comment (RC2) · Anonymous Referee #2 · 9 Aug 2016

**General comments**

The manuscript presents the results of a two-year comparison of total ozone measurements from Pandora, Brewer and Dobson spectrophotometers and satellite estimates (OMI, two algorithms). Results are given in terms of a least-squares analysis and some statistical indicators are provided (max, min, average, standard deviation, relative differences, Pearson's correlation index, $R^2$) together with the analysis of variance (ANOVA). The manuscript addresses relevant questions and could potentially provide interesting results, since a complete set of instrument operates at the measuring station of Seoul.

However, the manuscript is also disappointingly lacking of an appropriate analysis and looks just as a list of statistical indicators. Notably, a "Discussion" section examining in detail the sources of discrepancies is missing.

Indeed, it is stated that the differences between instruments (e.g., Pandora and Dobson) can be explained by a number of factors (e.g., SZAs, SO2, temperature, etc.), but no proof is provided. In a scientific paper, the authors should justify their statements based on a convincing analysis of a statistically representative number of data (e.g., SO2 measurements, effective ozone/instrumental temperatures, etc.). At least, the authors should provide a convincing proof that correcting for the above-mentioned factors of influence can improve the comparison and reduce the observed bias.

Also, the authors were not able to effectively disentangle the discrepancies arising from different sampling of datasets (on a daily or monthly basis) and those due to instrumental or processing issues.

Most of all, "very good agreement" among instruments is claimed, however differences are larger (up to 12.5%, cf. line 367) than the stated uncertainties (e.g., compare Figs. 5-6 and Sects. 1-2), therefore this issue should be commented in detail.

The comparison between ground-based instruments and the OMI spaceborne radiometer is also too vague. How much do temporal and spatial resolutions affect the comparison? How much aerosols and other pollutants impact on satellite estimates of total ozone over Seoul?

In conclusion, the manuscript should not be published, in my opinion, unless major revisions are introduced.

**Specific comments**

No mention is made in the manuscript of the processing/algorithms of the measurements. Do they differ for different instruments? For example, how much can the used spectroscopic sets of cross-sections (and their relative dependence on effective ozone

temperature) impact on the comparison?

Table 2: do the authors have an idea why some instruments have a negative intercept compared to Pandora and others a positive intercept?

How do the authors cope with straylight, which affects all data from the presented ground-based instruments? Lines 179-180 explain that measurements from Pandora at high SZAs are removed from the analysis, but what about the other instruments?

The comparison is performed in terms of monthly and daily averages. Then, a section (3.3) explains that large inconsistencies may originate due to the ozone variability when daily averages are performed on different datasets (because of different sampling frequencies). Could the authors additionally perform a comparison of nearly-simultaneous measurements? It is stated that the Pandora temporal resolution is about 2 minutes, therefore those measurements could be interpolated to the nearest Brewer/Dobson/OMI estimate, thus avoiding the bias illustrated in Sect. 3.3;

line 196: if the only condition for the comparison is that the number of daily observations must be greater than 10 days, it is likely that some differences are due to the sampled subset. Since the results orient the following comparison of the instruments, I think that the authors should revise their criteria or state the uncertainty of their results due to the day-to-day variability and different days sampled for each month in the datasets (as done in Sect. 3.3 for the daily averages);

The dependence on solar zenith angle is listed as an important factor impacting the comparison (line 365). Could the authors present some plots of the differences between instruments as a function of the solar zenith angle or the airmass?

What is the expected magnitude of the SO2 effect in Seoul (line 365)?

**Technical corrections**

- line 5 (and 209): it should be clarified why Pandora is taken as a reference for the comparison. Is it because of its high temporal resolution? In this case, this should be

explicitly stated;

- line 10: reporting the slope and $R^2$ for the comparison with the Brewer is redundant, since these data are already provided few lines above;

- line 12-13: are both instruments affected by these factors in a similar way?

- line 13: does "temperature" mean effective ozone temperature or instrumental temperature? Or both?

- Sect. 1: I would suggest to merge the description of the instruments in Sect. 1 with the text in Sect. 2. Instead, some references about previous comparisons of ozone instruments should be added in Sect. 1;

- line 25-27: please, add some bibliographic references;

- line 31: the recovery of the "ozone hole" is still an open question (e.g., Solomon et al., Science, 2016), and the cited articles (1997-2003) do not pretend to report the recovery of the ozone hole, contrary to what the authors state. Please, notice that slowdown of depletion does not necessarily mean recovery. Moreover, a quite confusing explanation of the "ozone hole" is offered to the reader (without even specifying where it occurs) and no mention to the decrease of ozone at midlatitudes is made. Please, rewrite this part;

- line 53: 363 nm. Is the Brewer an "extended" Brewer (MkIV-e)? If not, the readers might wonder why a single Brewer is able to measure at such large wavelengths;

- line 69: is the word "error" used in place of "uncertainty"?

- line 94: why "An" OMI?

- lines 121-127: I don't understand what this short review of previous investigations have to do with the aim of the manuscript. Please, focus on Brewer functioning and processing, instead;

- line 216: how can the authors state that Fig. 3 shows a "generally gaussian distribution" based on only 6 bins? Can they support their sentence on the basis of a normality test?

- line 221: "Dobson unit" or "Dobson instrument"?

- line 224: are you comparing Pandora, Brewer and OMI from 2012-2014 to the Dobson in the period 1985-2000? Please rewrite this sentence, since it is very confusing. Furthermore, can you assess an increase/decrease by comparing datasets from different instruments?

- lines 261-266: please, provide a short explanation about the additional information that can be extracted from the ANOVA analysis and the meaning of the ANOVA outputs that are presented in the text;

- lines 306-313: since the error sources are a key point of the comparison, this part should be commented in detail rather than just providing bibliographic citations;

- lines 306-309: what is the reason of the observed SZA-dependence?

- lines 312: "fixed temperature". Do you mean effective ozone temperature? Does humidity refer to instrumental humidity or atmospheric humidity?

- line 365: "temperature". Instrumental or ozone effective?

- Table 1: please, write more clearly the year (2012-2013) and the day;

- Fig. 4: define the used acronyms (OMT, OMD, DBS, etc.). Also, Fig. 4d has no dashed lines. Explain in the caption what the dashed lines indicate;

- Fig. 5, caption: "blue lines". There are no blue lines in the figures;

- Figs. 4-5: why n=115 for all plots? Only days with all instruments measuring were chosen? Is it necessary to report the same number in all plots?

[Figure]

---

## Short Comment (SC1) · 8 Sep 2016

Answers for Specific comments: line 21-22: The sentence is changed in line 20-23 line 49: The sentence is reformulated in line 50-51 line 115: a trend from 1979 to 2004 is divided into the former trend from 1979 to 1991 and the latter trend from 1992 to 2004 in line 115. In the section starting at line 118: It is mentioned as for the Dobson instrument, the calibrations (and their conclusions) that are relevant for the comparison period. line 179: The meaning of RMS and uncertainty is specified and reformulated in line 180. line 247-248: It means that all linear regression lines between Pandora and the others are close to 1 to 1 line. The sentence is changed. line 273: Yes MSR abbreviation for "mean square regression". line 303 and following: SO2 effect is mentioned in line 315. Figure 1 : It is hard to distinguish many TCO points from each instrument if the 5 panels are combined in one panel. Figure 3: Vertical axis is changed to "frequency" and its meaning is explained.

Please also note the supplement to this comment:
http://www.atmos-meas-tech-discuss.net/amt-2016-146/amt-2016-146-SC1-supplement.pdf

[Figure]

**Supplement:**

[revised manuscript text omitted]

detector, and 280–525 nm using the Sun-and-Sky CCD detector with absolute $O_3$ retrieval errors of about 1% ($\pm3$ DU) and a high precision of $\pm0.1$ DU (Herman et al., 2015; Reed et al.,

2015; Tzortziou et al., 2012). Absolute $NO_2$ retrieval errors are about $\pm0.1$ DU (Herman et al.,

2009). From the measured radiance, TCO levels, together with the total column of trace gases (including $NO_2$, $SO_2$, BrO, water vapor, and formaldehyde), are retrieved using the differential optical absorption spectroscopy (DOAS) technique (Wang et al., 2010; Yun et al.,

2013).

In this study, we intercompare the Pandora measurements from Seoul with two ground-based and two satellite datasets over a 2-year period. Furthermore, the difference between Pandora and the other measurements, and the causes of these differences, are discussed. The remainder of this paper is organized as follows. Section 2 describes the ground-based and satellite datasets used in this study. Section 3 describes the methodology and results of the intercomparison together with our analysis and discussion. In addition, high-resolution diurnal variations in the Pandora TCO data are compared with Dobson measurements. Finally, our conclusions are summarized in Sect. 4.

**2. Data and Analysis**

In this study, the TCO data used for intercomparisons were measured using Pandora, Dobson, and Brewer spectrophotometers from March 2012 to March 2014 at Yonsei University (37.57°N, 126.95°E; 84 m above sea level) in Seoul, Korea. The university is one of the WMO Global Ozone Observing System (GO3OS) stations (Station No. 252). OMI has also recorded TCO data over this site since 2004. As part of the ongoing national monitoring program of the Korea Meteorological Administration (KMA), TCO measurements have been made at this station since 1984. The calibration history and characteristics of Dobson (Beck #124), Brewer (SCI-TEC #148), and OMI instruments are described in Sect. 2.1 to 2.4.

**2.1. Dobson Spectrophotometer (Beck #124)**

The Dobson spectrophotometer (Beck #124) is located on the rooftop of the Science Hall of Yonsei University and has been in operation since 1984, with regular calibration as a standard for total ozone measurements (Cho, 1989, 1996; Cho et al., 2003; Kim et al., 2005). The instrument retrieves TCO from the observed UV radiance in direct-sun and zenith-sky modes three times a day. A direct-sun TCO value measured at noon under clear skies is generally selected as a representative value; however, a value close to noon or the zenith-sky measurement can be used instead if data from noon are unavailable. After the automation of the Dobson instrument (in particular, Q-levers, Attenuator, R-dial, observation, and data processing with test) in 2006, accuracy was improved such that the proportion of data points within a ±3% error range increased from 92% to 98% (Kim et al., 2007; Miyagawa et al., 2005). The calibration history of this instrument has been summarized by Kim et al. (2007)

and Hong et al. (2014). The Dobson instrument has provided a high-quality, objective, and reliable dataset that can be used to monitor the variations and trends in ozone levels over the

Korean Peninsula. According to previous studies that have used this dataset, the annual mean ozone level increased by 7.2% decade$^{-1}$ from 2004 to 2010 (Kim et al., 2014). This recent increase is contrary to the result reported in Kim et al. (2005) showing past decreasing trend of -2.39% decade$^{-1}$ from 1979 to 1991 measured by satellite TOMS, in spite of slight increasing trend of 0.75% decade$^{-1}$ from 1992 to 2004. As for the Dobson instrument, the calibrations are relevant for the comparison period in this result.

[revised manuscript text omitted]
 Dobson (a), Brewer (b), and OMI (TOMS (c) and DOAS (d)). Solid lines represent regression lines and dashed lines indicate an error range of ±3%.

[Figure]

**(a)**

**(c)**

**(b)**

**(d)**

**Figure 6.** Time series of relative differences in daily TCO values from Pandora and those from (a) Dobson, (b)
Brewer, (c) OMI-TOMS, and (d) OMI-DOAS ($\frac{TCO_{inst} - TCO_{Pan}}{TCO_{Pan}}$ [%]). Gaps in the time series indicate at least one
missing value in the observations from the four instruments.

[Figure]

**(a)**

**(c)**

**(e)**

**Figure 7.** Diurnal variations in TCO values retrieved from Pandora for six randomly selected clear-sky days
(cloud amount < 3) during the study period. TCO values measured at solar zenith angle > 75° are shaded and
were removed from the calculations. Filled circles and dashed lines represent direct-sun TCO values measured
by the Dobson instrument and observation times, respectively. All vertical axes have the same scale-range of
100 DU.

---

## Short Comment (SC2) · 8 Sep 2016

General comments: The manuscript presents the results of a two-year comparison of total ozone measurements from Pandora, Brewer and Dobson spectrophotometers and satellite estimates (OMI, two algorithms). Results are given in terms of a least-squares analysis and some statistical indicators are provided (max, min, average, standard deviation, relative differences, Pearson's correlation index, $R^2$) together with the analysis of variance (ANOVA). The manuscript addresses relevant questions and could potentially provide interesting results, since a complete set of instrument operates at the measuring station of Seoul.

ANSWER: We really appreciate the reviewer's all comments and suggestions about this manuscript. They are really helpful, so we can improve the details and quality of our manuscript based on these comments. The following content is the answers for the reviewer's comments and the revised manuscript is attached in supplement

Specific comments: 1. No mention is made in the manuscript of the processing/algorithms of the measurements. Do they differ for different instruments? For example, how much can the used spectroscopic sets of cross-sections (and their relative dependence on effective ozone C2 AMTD Interactive comment Printer-friendly version Discussion paper temperature) impact on the comparison?

ANSWER: General manuscripts of the processing/algorithms of four measurements are mentioned in Section 1. And the specific mention for each instrument in Seoul is summarized in Section 2.

2. Table 2: do the authors have an idea why some instruments have a negative intercept compared to Pandora and others a positive intercept? How do the authors cope with straylight, which affects all data from the presented ground-based instruments?

ANSWER: The intercept is associated with the slope. Actually, the intercept is relatively small when the slope value is large. Moreover, Pandora measurements are relatively small compared to other measurements (underestimation). Therefore, it is possible to find the negative value of the intercept if the difference is somewhat large at the small TCO. However in Table 2, all absolute values of t-ratio are in 0.27~1.14, so it is hard to say that the intercept is significantly negative. As for stray light, it affects all data especially in UV band, and its valid wavelength range is mentioned in Sabburg et al. (2002). In this study, Pandora, Brewer and Dobson data didn't use this wavelength range. Moreover, Dobson measures TCO using 2 pair wavelengths, so the instrument effect is small.

3. Lines 179-180 explain that measurements from Pandora at high SZAs are removed from the analysis, but what about the other instruments? The comparison is performed in terms of monthly and daily averages. Then, a section (3.3) explains that large inconsistencies may originate due to the ozone variability when daily averages are performed on different datasets (because of different sampling frequencies). Could the authors additionally perform a comparison of nearly simultaneous measurements? It is stated that the Pandora temporal resolution is about 2 minutes, therefore those measurements could be interpolated to the nearest Brewer/Dobson/OMI estimate, thus avoiding the bias illustrated in Sect. 3.3;

ANSWER: Basically, Dobson measures TCO under condition of mu < 2.5 as mentioned in Section 2.1 (noon/close to noon) and Brewer instrument is also set up to mu < 3.0. As mentioned in Section 3.3, large inconsistencies may generate due to the ozone variability, but the main purpose of this study is focused on the comparison of practical Pandora data (daily TCO) with other measurements rather than detailed validation as stated in line 371-374. Section 3.3 is the additional part in order to explain diurnal variation of TCO considered to the main cause of bias and Pandora data clearly shows this variation due to its high temporal resolution as mentioned in line 321. Validation of real-time data from each instrument leading to minimize bias is not the main purpose.

4. Line 196: if the only condition for the comparison is that the number of daily observations must be greater than 10 days, it is likely that some differences are due to the sampled subset. Since the results orient the following comparison of the instruments, I think that the authors should revise their criteria or state the uncertainty of their results due to the day-to-day variability and different days sampled for each month in the datasets (as done in Sect. 3.3 for the daily averages);

ANSWER: As mentioned above, the main purpose of this study is to compare practical Pandora data. So in spite of some differences due to sampling issue, daily Panodora TCO was calculated and the comparison result actually shows good agreement with Brewer, Dobson and OMI measurements, respectively. Also, monthly TCO values are

relatively less affected by daily variation and they are calculated when all data sets are available. The number of daily data is considerably small especially in summer season due to lack of clear days. Considering this, the comparison condition (daily observations > 10 per month) is set and monthly TCO values from all instruments are in good agreement with each other. The result of small mean relative difference stands for small day-to-day variability.

5. The dependence on solar zenith angle is listed as an important factor impacting the comparison (line 365). Could the authors present some plots of the differences between instruments as a function of the solar zenith angle or the airmass? What is the expected magnitude of the SO2 effect in Seoul (line 365)?

ANSWER: Two graphs in Figure 1 are generated from the comparison of Brewer and Dobson measurements in Seoul, Korea. The left figure shows the comparison of Brewer data and Dobson data after SO2 correction from 1999-2005 except for year 2001 in Seoul and the right figure shows the same comparison but from January 2007 to March 2007. From both figures, the agreements between the Dobson and Brewer dataset are reasonably good after SO2 correction with slopes of 0.996 (left) and 1.004 (right) respectively and R2 values of 0.986 (left) and 0.985 (right) respectively. Although there are some differences in data sampling, it can be known that slopes and R2 values in these figures are closer than those in Figure 4(e).

Technical corrections: 1. Line 5 (and 209): it should be clarified why Pandora is taken as a reference for the comparison. Is it because of its high temporal resolution? In this case, this should be C3 AMTD Interactive comment Printer-friendly version Discussion paper explicitly stated;

ANSWER: The reason Pandora data is taken as a reference for the comparison is that the operation of Pandora instrument was recently started compared to Dobson and Brewer in Seoul. So this study shows the reliability of Pandora data roughly through the result of inter-comparison with other instruments.

2. Line 10: reporting the slope and R2 for the comparison with the Brewer is redundant, since these data are already provided few lines above;

ANSWER: The redundant phrase is deleted.

3. Line 12-13: are both instruments affected by these factors in a similar way?

ANSWER: Line 12-13 states the difference between the Pandora and Dobson data is affected by these factors. The limited amount of data and SO2 effect generate the bias in Dobson and SZA dependence affects Pandora measurements. And the temperature and humidity affect both Dobson and Pandora measurements.

4. Line 13: does "temperature" mean effective ozone temperature or instrumental temperature? Or both?

ANSWER: It means Effective ozone temperature.

5. Line 25-27: please, add some bibliographic references;

ANSWER: Bibliographic references are added in manuscript.

6. Line 31: the recovery of the "ozone hole" is still an open question (e.g., Solomon et al., Science, 2016), and the cited articles (1997-2003) do not pretend to report the recovery of the ozone hole, contrary to what the authors state. Please, notice that slow-down of depletion does not necessarily mean recovery. Moreover, a quite confusing explanation of the "ozone hole" is offered to the reader (without even specifying where it occurs) and no mention to the decrease of ozone at midlatitudes is made. Please, rewrite this part;

ANSWER: This part is rewritten.

7. Line 94: why "An" OMI?

ANSWER: It is corrected to OMI

8. Line 216: how can the authors state that Fig. 3 shows a "generally gaussian distribution" based on only 6 bins? Can they support their sentence on the basis of a normality test?

ANSWER: The sentence is rewrited.

9. Line 221: "Dobson unit" or "Dobson instrument"?

ANSWER: It indicates Dobson instrument

10. Line 224: are you comparing Pandora, Brewer and OMI from 2012-2014 to the Dobson in the period 1985-2000? Please rewrite this sentence, since it is very confusing. Furthermore, can you assess an increase/decrease by comparing datasets from different instruments?

ANSWER: Line 224 is rewritten to clarify the sentence in manuscrpt.

11. Lines 312: "fixed temperature". Do you mean effective ozone temperature? Does humidity refer to instrumental humidity or atmospheric humidity?

12. Line 365: "temperature". Instrumental or ozone effective?

ANSWER: It refers to effective TCO retrieval temperature.

13. Table 1: please, write more clearly the year (2012-2013) and the day;

ANSWER: The sentence in Table 1 is revised to be more clearly..

14. Fig. 4: define the used acronyms (OMT, OMD, DBS, etc.). Also, Fig. 4d has no dashed lines. Explain in the caption what the dashed lines indicate;

ANSWER: Definition of acronyms and corrected line are shown in fig. 4.

15. Fig. 5, caption: "blue lines". There are no blue lines in the figures;

ANSWER: It changes to dashed lines in Fig. 5

16. Figs. 4-5: why n=115 for all plots? Only days with all instruments measuring were chosen? Is it necessary to report the same number in all plots?

ANSWER: In this comparison, only days with all instruments measuring were chosen to ensure high reliability of intercomparison results as mentioned in line 229-230.

Please also note the supplement to this comment:
http://www.atmos-meas-tech-discuss.net/amt-2016-146/amt-2016-146-SC2-supplement.pdf

————————————————

[Figure]

**Figure 1.** The comparison of Brewer and Dobson measurements in Seoul, Korea. The left figure shows the comparison of Brewer data and Dobson data after $SO_2$ correction from 1999-2005 except for year 2001 in Seoul. And the right figure shows the same comparison but from January 2007 to March 2007.

**Fig. 1.**

[Figure]

**Supplement:**

[revised manuscript text omitted]

detector, and 280–525 nm using the Sun-and-Sky CCD detector with absolute $O_3$ retrieval errors of about 1% ($\pm3$ DU) and a high precision of $\pm0.1$ DU (Herman et al., 2015; Reed et al.,

2015; Tzortziou et al., 2012). Absolute $NO_2$ retrieval errors are about $\pm0.1$ DU (Herman et al.,

2009). From the measured radiance, TCO levels, together with the total column of trace gases (including $NO_2$, $SO_2$, BrO, water vapor, and formaldehyde), are retrieved using the differential optical absorption spectroscopy (DOAS) technique (Wang et al., 2010; Yun et al.,

2013).

In this study, we intercompare the Pandora measurements from Seoul with two ground-based and two satellite datasets over a 2-year period. Furthermore, the difference between Pandora and the other measurements, and the causes of these differences, are discussed. The remainder of this paper is organized as follows. Section 2 describes the ground-based and satellite datasets used in this study. Section 3 describes the methodology and results of the intercomparison together with our analysis and discussion. In addition, high-resolution diurnal variations in the Pandora TCO data are compared with Dobson measurements. Finally, our conclusions are summarized in Sect. 4.

**2. Data and Analysis**

In this study, the TCO data used for intercomparisons were measured using Pandora, Dobson, and Brewer spectrophotometers from March 2012 to March 2014 at Yonsei University (37.57°N, 126.95°E; 84 m above sea level) in Seoul, Korea. The university is one of the WMO Global Ozone Observing System (GO3OS) stations (Station No. 252). OMI has also recorded TCO data over this site since 2004. As part of the ongoing national monitoring program of the Korea Meteorological Administration (KMA), TCO measurements have been made at this station since 1984. The calibration history and characteristics of Dobson (Beck #124), Brewer (SCI-TEC #148), and OMI instruments are described in Sect. 2.1 to 2.4.

**2.1. Dobson Spectrophotometer (Beck #124)**

The Dobson spectrophotometer (Beck #124) is located on the rooftop of the Science Hall of Yonsei University and has been in operation since 1984, with regular calibration as a standard for total ozone measurements (Cho, 1989, 1996; Cho et al., 2003; Kim et al., 2005). The instrument retrieves TCO from the observed UV radiance in direct-sun and zenith-sky modes three times a day. A direct-sun TCO value measured at noon under clear skies is generally selected as a representative value; however, a value close to noon or the zenith-sky measurement can be used instead if data from noon are unavailable. After the automation of the Dobson instrument (in particular, Q-levers, Attenuator, R-dial, observation, and data processing with test) in 2006, accuracy was improved such that the proportion of data points within a ±3% error range increased from 92% to 98% (Kim et al., 2007; Miyagawa et al., 2005). The calibration history of this instrument has been summarized by Kim et al. (2007)

and Hong et al. (2014). The Dobson instrument has provided a high-quality, objective, and reliable dataset that can be used to monitor the variations and trends in ozone levels over the

Korean Peninsula. According to previous studies that have used this dataset, the annual mean ozone level increased by 7.2% decade$^{-1}$ from 2004 to 2010 (Kim et al., 2014). This recent increase is contrary to the result reported in Kim et al. (2005) showing past decreasing trend of -2.39% decade$^{-1}$ from 1979 to 1991 measured by satellite TOMS, in spite of slight increasing trend of 0.75% decade$^{-1}$ from 1992 to 2004. As for the Dobson instrument, the calibrations are relevant for the comparison period in this result.

[revised manuscript text omitted]
 Dobson (a), Brewer (b), and OMI (TOMS (c) and DOAS (d)). Solid lines represent regression lines and dashed lines indicate an error range of ±3%.

[Figure]

**(a)**

**(c)**

**(b)**

**(d)**

**Figure 6.** Time series of relative differences in daily TCO values from Pandora and those from (a) Dobson, (b)
Brewer, (c) OMI-TOMS, and (d) OMI-DOAS ($\frac{TCO_{inst} - TCO_{Pan}}{TCO_{Pan}}$ [%]). Gaps in the time series indicate at least one
missing value in the observations from the four instruments.

[Figure]

**(a)**

**(c)**

**(e)**

**Figure 7.** Diurnal variations in TCO values retrieved from Pandora for six randomly selected clear-sky days
(cloud amount < 3) during the study period. TCO values measured at solar zenith angle > 75° are shaded and
were removed from the calculations. Filled circles and dashed lines represent direct-sun TCO values measured
by the Dobson instrument and observation times, respectively. All vertical axes have the same scale-range of
100 DU.

---

## Author Comment (AC1) · 27 Nov 2016

General comments: The paper compares the measurement of total ozone columns measured with the relatively new Pandora instrument with data from other techniques at a speciïňĄc site. Although similar comparisons have already been published it is of scientiïňĄc value to have this additional comparison to evaluate the quality of the data obtained with the different techniques. → We really appreciate the reviewer's all comments and suggestions about this manuscript. They are really helpful, thus helped improve the quality of our manuscript. The following is our responses to the reviewer's
comments and the revised manuscript is attached in supplement. Specific comments: line 21-22: "... are accurate and closely correlated" this is vague and could be more precise from the results. → "... are accurate and closely correlated" is changed more precisely to ", the daily TCO values measured by the Pandora during the 2-year study period proved its accuracy showing excellent correlation with R2 greater than 0.95 for other ground-based and satellite measurements". line 49: please reformulate with a separate sentence to make clear that the 1% accuracy holds for direct sun observations only → The sentence is reformulated to make clear that 1% accuracy holds for direct sun observations only as follows. "The accuracy of well calibrated Brewer measurements is estimated to be about 1% for direct-sun observations." line 115: a trend from 1979 to 2004 is mentioned while in line 102 it is mentioned that the instrument became operational in 1984. Please clarify → a trend from 1979 to 2004 is divided into the former trend from 1979 to 1991 measured by TOMS and the latter trend from 1992 to 2004 measured by Dobson in line 115. In the section starting at line 118: please mention, as for the Dobson instrument, the calibrations (and their conclusions) that is relevant for the comparison period. → The sentence was meant to the data quality of Dobson with regular calibration history which is described in the earlier part of the section with references. It is deleted to avoid confusion. line 179: please specify which RMS is meant (of the observations during the day?) and what is meant by uncertainty of ozone amount line 201-204: reformulate (split sentence). → root mean square (RMS) of weighted spectral fitting residuals < 0.05 (Here, RMS means the root mean square value of daily fitting residuals which are differences between fitting values and daily mean value). line 247-248: what is meant by "all () lines () show best fit"? → It means that all linear regression lines between Pandora and the others are close to 1 to 1 line. The sentence is changed to "That is, linear regression lines between Pandora and all others are very close to 1 to 1 line". line 273 : is MSR abbreviation for "mean square regression"? → Thanks. We added Mean Square Regression (MSR) line 303 and following: what would be the estimated effect of each of the possible error sources (eg what is the SO2 effect at Seoul?) → Typical concentration of SO2 in Seoul

is 0.02 ppm (annual average : 0.005 ppm, ∼1 DU when SO2 is assumed to distribute constantly up to 1 km in altitude), which affects O3 measurements but not significantly. Figure 1 : the 5 panels could be combined in one larger panel, with the different instruments represented in different colors/symbols. The same colors/symbols could then be used in Figure 2 → It would be hard to find each measurement clearly due to the large number of TCO points if the 5 panels are combined in one panel. Figure 3: if colors are used in Figs 1 and 2, the same colors can be used in fig 3. Please mention what is with of the bins of the histogram? → The name of vertical axis of Figure 3 is changed to "frequency" and it stands for the number of data in each TCO interval. Caption fig 6 (line 594): replace "null" by "missing" → "null" is replaced to "missing".

Please also note the supplement to this comment:
http://www.atmos-meas-tech-discuss.net/amt-2016-146/amt-2016-146-AC1-supplement.pdf

---

## Author Comment (AC2) · 27 Nov 2016

General comments: The manuscript presents the results of a two-year comparison of total ozone measurements from Pandora, Brewer and Dobson spectrophotometers and satellite estimates (OMI, two algorithms). Results are given in terms of a least-squares analysis and some statistical indicators are provided (max, min, average, standard deviation, relative differences, Pearson's correlation index, $R^2$) together with the analysis of variance (ANOVA). The manuscript addresses relevant questions and could potentially provide interesting results, since a complete set of instrument operates at the

measuring station of Seoul. → We really appreciate the reviewer's all comments and suggestions about this manuscript. They are really helpful, thus helped improve the quality of our manuscript. The following is our responses to the reviewer's comments and the revised manuscript is attached in supplement.

Specific comments: 1. No mention is made in the manuscript of the processing/algorithms of the measurements. Do they differ for different instruments? For example, how much can the used spectroscopic sets of cross-sections (and their relative dependence on effective ozone C2 AMTD Interactive comment Printer-friendly version Discussion paper temperature) impact on the comparison? → General manuscripts of the processing/algorithms of four measurements are mentioned in Section 1. And the specific mention for each instrument in Seoul is summarized in Section 2. 2. Table 2: do the authors have an idea why some instruments have a negative intercept compared to Pandora and others a positive intercept? How do the authors cope with straylight, which affects all data from the presented ground-based instruments? → The intercept represents bias of a measurement, associated with the slope of regression line. Pandora measurements tend to underestimate total column ozone(TCO) compared to other measurements. Therefore, it is possible to find the negative value of the intercept if the difference is somewhat large at the small TCO. However in Table 2, all absolute values of t-ratio are in 0.27∼1.14, so it is hard to say that the intercept is significantly negative. As for stray light, it affects all data especially in shorter UV band. Pandora corrects straylight empirically as a function of air mass factor(AMF), where the retrieved TCO decreases rapidly at very large air masses(sunrise and sunset) (Herman et al., AMT 2015). Sentences were added in Section 2.4. 3. Lines 179-180 explain that measurements from Pandora at high SZAs are removed from the analysis, but what about the other instruments? The comparison is performed in terms of monthly and daily averages. Then, a section (3.3) explains that large inconsistencies may originate due to the ozone variability when daily averages are performed on different datasets (because of different sampling frequencies). Could the authors additionally perform a comparison of nearly simultaneous measurements? It is stated that

the Pandora temporal resolution is about 2 minutes, therefore those measurements could be interpolated to the nearest Brewer/Dobson/OMI estimate, thus avoiding the bias illustrated in Sect. 3.3; → Basically, Dobson measures TCO under condition of mu < 2.5 as mentioned in Section 2.1 (noon/close to noon) and Brewer instrument is also set up to mu < 3.0. As mentioned in Section 3.3, large inconsistencies may generate due to the ozone variability, but the main purpose of this study is focused on the comparison of daily TCO by Pandora, with other measurements as stated. Section 3.3 is the additional part in order to explain diurnal variation of TCO and Pandora data clearly shows this variation with its high temporal resolution as mentioned in line 321. Validation of real-time data from each instrument leading to minimize bias has been done in previous studies and is not the main focus of current study. 4. Line 196: if the only condition for the comparison is that the number of daily observations must be greater than 10 days, it is likely that some differences are due to the sampled subset. Since the results orient the following comparison of the instruments, I think that the authors should revise their criteria or state the uncertainty of their results due to the day-to-day variability and different days sampled for each month in the datasets (as done in Sect. 3.3 for the daily averages); → As mentioned above, the main objective of this study is to compare Pandora data with other instruments in terms of daily values. Thus, despite of some differences in sampling issue, daily Panodora TCO was calculated and the comparison result actually shows good agreement with Brewer, Dobson and OMI measurements, respectively. Also, monthly TCO values are relatively less affected by daily variation and they are calculated when all data sets are available. The number of daily data is considerably small especially in summer season due to lack of clear days. Considering this, the comparison condition (daily observations > 10 per month) is set and monthly TCO values from all instruments are in good agreement with each other. The result of small mean relative difference stands for small day-to-day variability. 5. The dependence on solar zenith angle is listed as an important factor impacting the comparison (line 365). Could the authors present some plots of the differences between instruments as a function of the solar zenith angle or the airmass? What is

the expected magnitude of the SO2 effect in Seoul (line 365)? → Unfortunately, other instruments except for Brewer have taken measurements one to maximum three times per day, which limits the direct comparison of SZA effect. Two graphs in Figure 1 are generated from the comparison of Brewer and Dobson measurements in Seoul, Korea. The left figure shows the comparison of Brewer data and Dobson data after SO2 correction from 1999-2005 except for year 2001 in Seoul and the right figure shows the same comparison but from January 2007 to March 2007. From both figures, the agreements between the Dobson and Brewer dataset are reasonably good after SO2 correction with slopes of 0.996 (left) and 1.004 (right) respectively and R2 values of 0.986 (left) and 0.985 (right) respectively. Typical concentration of SO2 in Seoul is 0.02 ppm (annual average : 0.005 ppm, ∼1 DU when SO2 is assumed to distribute constantly up to 1 km in altitude), which affects O3 measurements but not significantly. Although there are some differences in data sampling, it can be known that slopes and R2 values in these figures are closer than those in Figure 4(e).

Technical corrections: 1. Line 5 (and 209): it should be clarified why Pandora is taken as a reference for the comparison. Is it because of its high temporal resolution? In this case, this should be C3 AMTD Interactive comment Printer-friendly version Discussion paper explicitly stated; → Sorry for the confusion. Pandora is not a reference in this study. Rather, it meant to compare the recent Pandora measurement with other collocated instruments such as Dobson and Brewer in Seoul, in addition to OMI satellite data. So this study attempts to assess the reliability of Pandora data in terms of daily representative values through the result of inter-comparison with other instruments. 2. Line 10: reporting the slope and R2 for the comparison with the Brewer is redundant, since these data are already provided few lines above; → The redundant phrase is deleted. "In particular, they show a close agreement with the Brewer TCO measurements. . ." 3. Line 12-13: are both instruments affected by these factors in a similar way? → Line 12-13 lists the possible explanation for the difference between the Pandora and Dobson data is affected by these factors. The way the two instruments are affected by those factors except for SO2 are similar in qualitatively. Pandora

retrieves ozone amounts from spectral fitting to cover the entire 310 to 330 nm range, while Dobson retrieves ozone amounts from the difference between the intensity of selected wavelength pairs in the range 300-340 nm. Sentences were added in Section 2.4. 4. Line 13: does "temperature" mean effective ozone temperature or instrumental temperature? Or both? → Yes, it means "Effective Ozone temperature". Thanks. Corrected. 5. Line 25-27: please, add some bibliographic references; → Bibliographic references are added in manuscript (Liou, 2002; Schott, 2007). 6. Line 31: the recovery of the "ozone hole" is still an open question (e.g., Solomon et al., Science, 2016), and the cited articles (1997-2003) do not pretend to report the recovery of the ozone hole, contrary to what the authors state. Please, notice that slowdown of depletion does not necessarily mean recovery. Moreover, a quite confusing explanation of the "ozone hole" is offered to the reader (without even specifying where it occurs) and no mention to the decrease of ozone at midlatitudes is made. Please, rewrite this part; → This part is rewritten. 7. Line 94: why "An" OMI? → It is corrected to OMI 8. Line 216: how can the authors state that Fig. 3 shows a "generally gaussian distribution" based on only 6 bins? Can they support their sentence on the basis of a normality test? → Lin3 216 is rewritten. 9. Line 221: "Dobson unit" or "Dobson instrument"? → It means Dobson instrument 10. Line 224: are you comparing Pandora, Brewer and OMI from 2012-2014 to the Dobson in the period 1985-2000? Please rewrite this sentence, since it is very confusing. Furthermore, can you assess an increase/decrease by comparing datasets from different instruments? → Sorry for the confusion. It meant to compare the current study period of 2012-2014 to previous data record. Sentences were rewritten and deleted to avoid confusion. 11. Lines 312: "fixed temperature". Do you mean effective ozone temperature? Does humidity refer to instrumental humidity or atmospheric humidity? → Effective temperature used in TCO retrieval. Humidity was meant to be the instrumental humidity due to humidity outside. 12. Line 365: "temperature". Instrumental or ozone effective? → It refers to effective temperature for TCO retrieval. 13. Table 1: please, write more clearly the year (2012-2013) and the day; → The title in Table 1 states the period of

measurements clearly.. 14. Fig. 4: define the used acronyms (OMT, OMD, DBS, etc.). Also, Fig. 4d has no dashed lines. Explain in the caption what the dashed lines indicate; → Definition of acronyms and corrected line are shown in fig. 4 and its title. 15. Fig. 5, caption: "blue lines". There are no blue lines in the figures; → It now changed to dashed lines in Fig. 5 16. Figs. 4-5: why n=115 for all plots? Only days with all instruments measuring were chosen? Is it necessary to report the same number in all plots? → In this comparison, only days with all instruments measuring were chosen to have fair and reliable intercomparison as mentioned in line 229-231.

Please also note the supplement to this comment:
http://www.atmos-meas-tech-discuss.net/amt-2016-146/amt-2016-146-AC2-supplement.pdf